# Systematic Identification and Functional Analysis of the *Hypericum perforatum* L. bZIP Gene Family Indicating That Overexpressed *HpbZIP69* Enhances Drought Resistance

**DOI:** 10.3390/ijms241814238

**Published:** 2023-09-18

**Authors:** Ruyi Xiao, Yan Sun, Shu Yang, Yixiao Yang, Donghao Wang, Zhezhi Wang, Wen Zhou

**Affiliations:** Key Laboratory of the Ministry of Education for Medicinal Resources and Natural Pharmaceutical Chemistry, National Engineering Laboratory for Resource Development of Endangered Crude Drugs in Northwest of China, Shaanxi Normal University, Xi’an 710119, China; xiaoruyi@snnu.edu.cn (R.X.); 15319720188@163.com (Y.S.); yangshu113@xab.ac.cn (S.Y.); yixiao@snnu.edu.cn (Y.Y.); wangdonghao@snnu.edu.cn (D.W.)

**Keywords:** bZIP transcription factor, *Hypericum perforatum*, drought stress

## Abstract

Basic leucine zipper (bZIP) transcription factors play significant roles in plants’ growth and development processes, as well as in response to biological and abiotic stresses. *Hypericum perforatum* is one of the world’s top three best-selling herbal medicines, mainly used to treat depression. However, there has been no systematic identification or functional analysis of the bZIP gene family in *H. perforatum*. In this study, 79 *HpbZIP* genes were identified. Based on phylogenetic analysis, the *HpbZIP* gene family was divided into ten groups, designated A–I and S. The physicochemical properties, gene structures, protein conserved motifs, and Gene Ontology enrichments of all *HpbZIPs* were systematically analyzed. The expression patterns of all genes in different tissues of *H. perforatum* (i.e., root, stem, leaf, and flower) were analyzed by qRT-PCR, revealing the different expression patterns of *HpbZIP* under abiotic stresses. The HpbZIP69 protein is localized in the nucleus. According to the results of the yeast one-hybrid (Y1H) assays, HpbZIP69 can bind to the *HpASMT2* (N-acetylserotonin *O*-methyltransferase) gene promoter (G-box *cis*-element) to activate its activity. Overexpressing *HpbZIP69* in *Arabidopsis* wild-type lines enhanced their tolerance to drought. The MDA and H_2_O_2_ contents were significantly decreased, and the activity of superoxide dismutase (SOD) was considerably increased under the drought stress. These results may aid in additional functional studies of *HpbZIP* transcription factors, and in cultivating drought-resistant medicinal plants.

## 1. Introduction

Basic leucine zipper (*bZIP*) TFs are a widely distributed and highly conserved multigene family in eukaryotes [1]. They play an essential role in regulating plants’ growth and development processes and promoting the synthesis of secondary metabolites. The bZIP protein is famous for its highly conserved bZIP domains, consisting of 40~60 amino acids [2]. This domain consists of two structural features: one is the primary binding region (N-X7-R/K-X9), and the other is the leucine zipper dimer domain [3]. The basic binding region consists of about 18 amino acid residues and binds to specific DNA sequences with the help of a fixed N-X7-R/K structure [4]. The leucine zipper dimer domain is composed of several heptavalent repeats of leucine or other hydrophobic amino acids, which bind tightly to the essential region. The leucine zipper forms an amphiphilic α-helix, which can regulate the homology or heterodimerization of the bZIP protein before binding to DNA [5]. The bZIP proteins in plants have binding specificity for DNA sequences containing ACGT elements, which promote preferential binding to G-box (CACGTG), C-box (GACGTC), and A-box (TACGTA) [6]. Upon DNA binding, one-half of the N-terminal basic binding region inserts into the large groove of double-stranded DNA. The remaining half of the C-terminal leucine zipper regulates dimerization, thus forming a superimposed helical coil structure [7]. Such a structure also determines that bZIP transcription factors can regulate the expression of multiple downstream genes by interacting with cis-acting elements in the promoter region, thus participating in the transcriptional regulation process.

At present, a variety of *bZIP* transcription factors have been shown to regulate the transcriptional expression of related genes by interacting with genes in response to biotic and abiotic stresses [8]. Under salt stress conditions, *AtbZIP17* in *Arabidopsis* directly or indirectly regulates some salt-stress-related response genes, thereby participating in the salt signaling cascade [9]. *AtbZIP53* can interact with *AtbZIP10* to form dimers, further regulate proline metabolism, and participate in abiotic stress response [10]. *AtbZIP53* can also bind to *AtbZIP1*, participate in carbon and nitrogen metabolism pathways, and affect the decomposition and metabolism of sugars and amino acids [11]. In the study of *Brachypodium distachyon* and *Oryza sativa*, it was found that the expression levels of *BdbZIP30* and *BdbZIP41*, along with their homologous genes *OsbZIP63* and *OsbZIP05* in *O. sativa*, increased under high-salt conditions [12,13]. Overexpression of the wheat (*Triticum aestivum*) gene *TabZIP6* in *Arabidopsis* reduced its frost resistance [14]. Overexpression of *Camellia sinensis bZIP6* can enhance the ability to resist low-temperature stress in *Arabidopsis* [15]. Furthermore, *bZIP* transcription factors can specifically bind to ABREs (abscisic-acid-responsive elements) in response to drought stresses [16,17]. For example, *AtbZIP36* can be combined with the cis-acting ABRE in the promoter region to promote and inhibit the expression of stress-regulatory proteins and participate in the stress response pathway, thus improving the drought tolerance of *Arabidopsis* [18]. Moreover, ramie *BnbZIP2* was overexpressed in *Arabidopsis*, which made transgenic *Arabidopsis* show stronger drought resistance compared with the wild type [19]. The exogenous hormone ABA significantly affects *bZIP* transcription factors. In the study of *TabZIP14-B* in wheat, it was found that *TabZIP14-B* transgenic plants were more sensitive to ABA than wild types, which severely inhibited the plants’ root growth [20]. *OsbZIP62*, a stress-responsive *bZIP* transcription factor, was found to improve drought resistance and oxidative tolerance in rice [21]. On the basis of expression pattern analyses, *JcbZIP49* and *JcbZIP50* are likely involved in responses to drought stress in *Jatropha curcas* [22]. The overexpression of *TabZIP8-7A* conferred greater drought resistance and ABA sensitivity in *Arabidopsis* [23]. Transcript accumulation of *AtbZIP62* and *AtPYD1* showed that both were highly upregulated by drought stress in wild-type (WT) plants [24]. The TGA subfamily of *bZIP* transcription factors has been studied in depth, proving that this family is relevant to plants’ disease resistance [4,25,26]. For instance, TGA2, TGA5, and TGA6 are essential for plants’ disease resistance and have redundant functions [27].

The planting area for traditional Chinese medicine is constantly increasing in China, and the demand for land is also increasing. Promoting the cultivation of *H. perforatum* in arid and semi-arid areas has become one of the important means to alleviate its resource shortage. Drought is a highly destructive and frequent global natural disaster that restricts the yield and quality of traditional Chinese medicinal materials. Therefore, studying stress-resistance genes and improving the ability of *H. perforatum* to resist drought stress is of great significance for improving its yield and quality [28]. Numerous studies have shown that the bZIP transcription factors have many important biological functions, particularly in enhancing plants’ drought resistance [29,30]. To reveal the detail and to facilitate future research on the *bZIP* TF family in *H. perforatum*, the gene figure, classification, and stress-produced expression modes of bZIP TF family members in *H. perforatum* were systematically analyzed based on whole-genome data. The phenotype and RNA-Seq information of *HpbZIP69* transgenic *Arabidopsis* were analyzed to investigate the characteristics and molecular functions of *HpbZIP69* under drought stress. The outcomes will lay a foundation for the molecular biology of drought resistance in *H. perforatum.*

## 2. Results

### 2.1. Identification and Sequence Features

The *Arabidopsis* bZIPs were taken as the comparison objects. Through screening out of the *H. perforatum* genome database (No. PRJNA588586), 79 members of the *H. perforatum* bZIPs were finally obtained and named as HpbZIP1 to HpbZIP79. Obtained via bioinformatics software analysis, the physicochemical properties of the bZIPs of *H. perforatum* are shown in Appendix A. The lengths of the proteins encoded by the *H. perforatum* bZIPs range from 119 aa (HpbZIP46) to 575 aa (HpbZIP79); the isoelectric points range from 4.65 (HpbZIP21) to 10.02 (HpbZIP53); the molecular weights range from 14.27 KDa (HpbZIP46) to 73.8 KDa (HpbZIP32). The subcellular localization of all genes was predicted, and the results are shown in Appendix A.

### 2.2. Phylogenetic Analysis of HpbZIPs

To seek the evolutionary relationships of bZIP TFs, we generated a phylogenetic tree, including 75 *Arabidopsis* and 79 *H. perforatum* bZIPs, based on the alignment of the amino acid sequences (Figure 1). Based on the topology of the phylogenetic tree and the category of *Arabidopsis* bZIPs, ten groups (named A–I and S) were categorized. According to the distribution characteristics of conserved domains and other structural components in its protein structure, the *H. perforatum* bZIP gene family can be divided into ten corresponding groups. The structural diagram of each subfamily representing the protein is shown in Appendix A. Moreover, HpbZIP5 and HpbZIP36 were not classified into any group, which indicates that the sequences have varied in the process of evolution, and the functions may have changed considerably. Based on the phylogenetic tree, the largest cluster is Group S, containing 20 *AtbZIP*s and 19 *HpbZIP*s. The second-largest cluster is Group A, which contains 12 *AtbZIP*s and 14 *HpbZIP*s. According to the gene functions of different groups, Group A is mainly involved in ABA and stress-mediated signal transduction. In contrast, Group S is mainly involved in biological processes in response to stresses, such as cold, drought, and injury. The members of Group S and Group A are almost all involved in drought and ABA stress responses, further demonstrating that Group S and Group A play essential roles in the stress regulation of plants. Unlike *Arabidopsis*, Group S and Group I can be subdivided into four subgroups (designated as Sa, Sb, Ia, and Ib) due to minor differences in gene clustering.

### 2.3. Gene Structure and Conserved Motif Analyses

To further research the phylogenetic connections among *HpbZIP* TFs, an unrooted phylogenetic tree containing only *HpbZIP*s was made, and the exon–intron structure and allocations of conserved domains in *HpbZIP*s were analyzed (Figure 2). The *HpbZIP*s were classified and named as above. The results of multiple sequence alignment forecasted that every *HpbZIP* had a conserved bZIP domain, and its *N*-terminal region was an alkaline region containing an N-X7-R/K motif, as well as its C-terminal leucine zipper region containing leucine every seven hydrophobic amino acids. The exon–intron structures of *HpbZIPs* between members are shown in Figure 2. The genetic structures suggested significant specificity between groups. Because the same subtribe in introns and exons is relatively similar, the distribution and quantity of splice sites were fairly conservative, the same subtribe genes had closer evolutionary relationships. Some *HpbZIP* genes (28%) were intronless. Six genes only had CDS sequences: *HpbZIP18*, *HpbZIP13*, *HpbZIP53*, *HpbZIP63*, *HpbZIP15*, and *HpbZIP68*. The number of exons was 12 at most and 1 at least. The number of introns was 11 at most and 1 at least. Compared with other genes in the bZIP gene family, the length of *HpbZIP46* was the largest, at about 5–6 times the average length of other genes, with very long introns and very short CDS sequences. There were seven groups (*HpbZIP6/HpbZIP7*, *HpbZIP31/HpbZIP41*, *HpbZIP17/HpbZIP67*, *HpbZIP50/HpbZIP79*, *HpbZIP38/HpbZIP74*, *HpbZIP19/HpbZIP51*, and *HpbZIP57/HpbZIP62*) that contained the same number of exons and of similar length, suggesting that there may be tandem repeats in the *HpbZIP* gene family.

We identified 15 conserved motifs in 79 HpbZIP proteins (Appendix A) based on the phylogenetic tree and MEME online program. Different bZIP genes have different types of conserved motifs, and the number of motifs differs, suggesting different potential functional sites and different biological functions that may be involved. Although different bZIP genes have different types of conserved motifs, the type and number of conserved motifs in the same subfamily are still relatively similar. The length of all motifs was between 6 and 50 amino acids. Some of the bZIP members had six motifs; other bZIP members owned at least two. Motif 1 existed in almost every HpbZIP. Some motifs existed in various genes. For example, motif 14 was identified in 50 HpbZIPs. A few motifs, like motif 13, only existed in four HpbZIPs, while most conserved motifs occurred in particular groups. For instance, motif 9 and motif 3 were found in Group B, motif 8 and motif 2 were identified in Group D, and motif 5 and motif 12 existed in Group C (Appendix A). The similar composition and distribution patterns of exon–intron structures and conserved domains facilitated the phylogenetic connection and categorization of the *HpbZIP* TFs.

### 2.4. Transcript Abundance Profiling

The upstream 1.5 kb of ATG was used to predict cis-acting elements. As shown in Appendix A, plenty of cis-acting elements connected with growth and development, hormonal regulations, and stresses were identified in 79 *HpbZIP* gene promoters. There were 64 genes responding to drought, including DRE and MBS elements, and 176 involved in photoresponse regulation, such as GT1-motif, MNF1, Box I, 3-AF1 binding site, ATCC motif, 4cl-CMA2b, and 45 contributing to abscisic acid response (ABREs). Analysis of cis-acting elements of *HpbZIP* gene family members, and especially the functional annotation, contributed to the subsequent study of available *bZIP* genes. For the GO annotation analysis, 74 HpbZIP proteins were summarized into 25 functional subcategories (Appendix A) of the three main ontologies according to amino acid similarity. There were only four proteins enriched in a membrane, membrane part, organelle part, and protein-containing complex. The number of proteins enriched in a cell, cell part, and organelle was 74. Among the molecular functions, the proteins enriched in the GO annotation on the nutrient storage activity were the minimum, and the proteins enriched in the binding and transcription-regulatory activity were the most intensive. In addition, some proteins were increased in response to stimuli and biological regulation processes, accounting for about 50% of the total protein number. It is speculated that members of this protein family can bind cis-acting elements to activate the expression of critical genes in the transcriptional regulation process.

### 2.5. Analysis of Expression Patterns 

Our laboratory has established the transcriptome database of *H. perforatum* in different tissues (flower, leaf, root, and stem; SRR8438983-SRR8438986). The formula log_10_(FPKM) from the RNA-Seq data was applied to hierarchical clustering. The *HpbZIP* genes displayed different expression patterns in the four organs shown in the heatmap (Figure 3). For example, the expression of *HpbZIP59*, *HpbZIP70*, and *HpbZIP77* was similar in all tissues. However, some genes were highly expressed in specific sites and low in others, such as *HpbZIP11*, *HpbZIP15*, and *HpbZIP27*, implying that gene expression might be development-specific. 

In addition, in order to investigate the response of members of the *HpbZIP* gene family to abiotic stresses, we selected 1–2 representative genes from 10 subfamilies according to the results of the evolutionary groups, and we analyzed the expression patterns of 12 genes under hormone treatment (ABA; Figure 4A) and abiotic stresses (NaCl, PEG; Figure 4B,C). RT-qPCR was used to detect the expression levels at 0, 1, 3, 6, and 12 h. The expression levels of different genes were increased under ABA treatment. With the increase in treatment time, *HpbZIP14*, *HpbZIP69*, and *HpbZIP37* showed a significant upregulation trend, while *HpbZIP31* and *HpbZIP40* showed a downregulation trend. The expression of *HpbZIP31* increased sharply at a specific timepoint. Under drought and high-salt treatment, gene expression levels were generally higher, and the expression levels showed a more apparent temporal difference (Figure 4B,C). Among them, the relative expression levels of *HpbZIP69* were significantly increased under drought treatment compared with the control (0 h). The numerous expression patterns revealed the different roles of *HpbZIP* genes in abiotic stress response pathways.

### 2.6. Characterization of Transcription Activity of HpbZIP69

According to the evolutionary relationship, *HpbZIP69* is homologous to *ABF3* (*AtbZIP37*) and *ABF4/AREB2* (*AtbZIP38*) [18,31], which are involved in the regulation of abiotic stresses such as drought. It is speculated that *HpbZIP69* also has a similar function. Meanwhile, combined with the analysis of gene expression patterns under drought treatment, the expression level of *HpbZIP69* was also significantly increased. The full length of *HpbZIP69* cDNA is 1125 bp, encoding 374 amino acids, with a molecular weight of 40.28 KDa and an IP of 9.4. Bioinformatics software analysis showed that HpbZIP69 was mainly localized in the nucleus (Appendix A), which is consistent with the general characteristics of transcription factors. To verify the localization of HpbZIP69, the *HpbZIP69* was fused with GFP driven by the 35S promoter and transiently expressed in onion epidermal cells. The *HpbZIP69*-GFP was detected in the nucleus (Figure 5A), indicating that HpbZIP69 is a nucleus-localized protein. As shown in Figure 5B, the *N*-terminal region (1-298 aa) without the bZIP domain had minimal demand for its trans activity, as bZIP transcription factors bound the G-box motif (CACGTG). The yeast one-hybrid assay indeed showed that HpbZIP69 binds G-box with a conserved sequence (Figure 5C). The abovementioned results show that *HpbZIP69* is a characteristic bZIP transcription factor.

### 2.7. Overexpression of HpbZIP69 in Arabidopsis

To investigate the function of *HpbZIP69*, pEarleyGate202-*HpbZIP69* was transferred into *Arabidopsis* (Col-0). The target gene was amplified and identified to verify whether the transgenic plants were positive. *HpbZIP69* was cloned from six candidate transgenic lines, but no bands were found at 1000–2000 bp in the WT (Figure 6A). Three independent stable homozygous lines were acquired. They were confirmed using semi-qRT-PCR (Figure 6B). According to the results of semi-qRT-PCR, the expression of *HpbZIP69* was higher in OE-1 than in OE-2 and OE-3, so the OE-1 line was selected for subsequent functional assays. To see whether *HpbZIP69* can enhance the drought resistance of plants, one-month-old field *Arabidopsis* seedlings were used for drought tolerance analysis. The growth development of the OE lines was obviously better than that of the WT without water for 15 days (Figure 6C). The contents of SOD in the OE lines were higher than that in the WT under drought treatment, and the MDA and H_2_O_2_ contents in the OE lines were notably lower than in the WT (Figure 6D). The aforementioned results revealed that the ectopic expression of *HpbZIP69* enhanced the drought tolerance in *Arabidopsis*.

## 3. Discussion

The *H. perforatum* bZIP gene family is a multigene family that is widely present and highly conserved in eukaryotes, where it plays a vital role in regulating plants’ growth and development, participating in stress response, and promoting the synthesis of secondary metabolites [32]. In this study, based on a genome-wide database of *H. perforatum*, we screened 79 *HpbZIP* genes and the relationships between their physical and chemical properties, system development evolution, gene structures, conserved motifs, gene function enrichment regions, and cis-acting elements, as well as under different tissues and different hormone gene expression patterns, and we conducted a preliminary analysis. Phylogenetic analysis showed that the bZIP gene family of *H. perforatum* had the same evolutionary group as that of *A. thaliana*, which was also divided into ten groups, among which group A was mainly responsible for transcriptional regulation and various stress responses [33]. 

According to the clustering results of the evolutionary tree, *HpbZIP69* was closely clustered with *ABF3* (*AtbZIP37*) and *ABF4/AREB2* (*AtbZIP38*), which is involved in the regulation of abiotic stresses such as drought. Therefore, we speculate that the two may have similar functions. In this study, tissue culture seedlings of *H. perforatum* at two months of age were subjected to drought. The qRT-PCR results showed that the expression levels of *HpbZIP69* were significantly increased under drought treatment, suggesting that it may be involved in regulating drought stress. *HpbZIP69* has cis-acting elements that respond to drought, abscisic acid, and excessive salt stimulation, which can promote transcriptional activation of genes related to the stress response. Therefore, the expression patterns of *HpbZIP69* under drought, abscisic acid, and high-salt treatments were analyzed simultaneously. The results showed that the gene expression levels of *HpbZIP69* increased after these treatments at different times, but the gene expression levels were different in time and space. We also constructed the pEarleyGate103 (CD3-685) subcellular localization vector. We found that HpbZIP69 was only expressed in the nucleus through transient expression of the target gene in onion with the help of gene gun transformation technology. According to the results of transcriptional self-activation and yeast single-hybridization experiments, we found that *HpbZIP69* has self-activation activity and can interact with the G-box motif.

In this investigation, *HpbZIP69* was found to be resistant to drought stress to a certain extent, but the mechanism through which it responds to drought stress remains to be further studied. It has been reported that Group A members in *A. thaliana*, such as *ABF3* (*AtbZIP37*) and *ABF4/AREB2* (*AtbZIP38*), have strong transcriptional regulation ability. Meanwhile, they can control the transcriptional activity of essential enzyme genes in the drought response pathway by binding cis-acting elements on promotors, participating in the stress response process. Analysis of cis-acting elements in this study showed that the cis-acting element AREB was closely related to stress regulation. Subsequent experiments could explore whether the HpbZIP69 transcription factors bind to promoters and pertain to ABA-mediated transcriptional activation. In addition, ABA-mediated transcriptional activity is related to flavonoid synthesis [34], so whether *HpbZIP69* plays a role in the flavonoid metabolism pathway could be explored, and the drought response mechanism of *HpbZIP69* could be further elucidated.

Previous studies have reported that ABF3 (AtbZIP37) and ABF4/AREB2 (AtbZIP38) can regulate the transcriptional activation of essential enzyme genes in the stress response pathway in *A. thaliana* [18,31], thereby contributing to the stress response. Based on the phylogenetic analysis of *H. perforatum* and *A. thaliana*, it was found that *HpbZIP69* was closely clustered with *AtbZIP37/38*, suggesting that it may have a similar function in response to drought. In this study, wild-type and overexpressing transgenic *Arabidopsis* lines were subjected to both control conditions and drought treatments. Phenotypic analysis showed that the overall growth of *OE-HpbZIP69* lines was better than that of the wild type. We also determined the physiological indices related to drought resistance. The results showed that the MDA [35], H_2_O_2_ [36], and reactive oxygen species contents of OE-*HpbZIP69* lines were significantly lower than those of the wild type, and the activity of superoxide dismutase (SOD) [37] was considerably higher than that of the wild type under drought stress. Based on the above results, we believe that *HpbZIP69* can enhance the drought resistance of plants, suggesting that it plays an influential role in regulating plants’ drought resistance.

## 4. Materials and Methods

### 4.1. Identification and Sequence Analysis

The typical protein sequences of *Arabidopsis* and rice downloaded from TAIR (http://www.arabidopsis.org/, accessed on 20 October 2022) [38] and RGAP (http://rice.plantbiology.msu.edu/, accessed on 20 October 2022) were used as a query to probe the *H. perforatum* genome assembly in our lab, using hmmer3.1 [39,40] and Pfam (http://pfam.sanger.ac.uk/, accessed on 20 October 2022) [41]. The E value was set to be 1 during the comparisons. The initial sequences were searched in Conserved Domains Search (https://www.ncbi.nlm.nih.gov/Structure/cdd/wrpsb.cgi, accessed on 5 November 2022) [42]. After deleting repeated and incomplete gene sequences, the remaining ones containing bZIP domains were considered to be *H. perforatum bZIP* family members. The arrangements were named in order to screen them out from the library. Every HpbZIP protein sequence was uploaded to ExPASy (https://web.expasy.org/protparam/, accessed on 7 November 2022) [43] to determine the amino acid quantity, molecular weight, and theoretical isoelectric point. The conserved motifs in bZIP proteins were analyzed utilizing MEME (https://meme-suite.org/meme/tools/meme, accessed on 10 November 2022) [44]. The E values of motifs less than 1 × 10^−10^ were kept for subsequent analysis.

### 4.2. Phylogenetic Analysis, Cis-Acting Element Predictions, and Gene Ontology Annotations

Protein sequences of bZIPs from *H. perforatum* and *Arabidopsis* were used for phylogenetic analysis, where they were aligned using Clustal X2.0.8. The unrooted phylogenetic trees were generated using MEGA 7.0 with the neighbor-joining method. In the parameter settings, the gaps/missing data treatment was set to pairwise deletion, and the model developed was a Poisson model. The bootstrap test was run 1000 times, and the other parameters were checked against default values. To predict and classify the promoter cis-acting element composition, the 1.5 kb upstream sequence of each *HpbZIP* from ATG was uploaded to BioEdit and PlantCARE (http://bioinformatics.psd.ugent.be/webtools/plantcare/html/, accessed on 10 November 2022) [45]. The protein sequences were matched to the NCBI non-redundant protein database with Blast2GO [46] to acquire a Gene Ontology (GO) annotation for each of the *HpbZIPs*. Furthermore, WEGO 2.0 [47] was applied to generate GO functional classifications and allocate gene functions at the macro level.

### 4.3. Plant Materials, Stress Treatments, and Expression Analysis

Seeds of *H. perforatum* were purchased from Gansu Province. The seeding methods and cultivation conditions of two-month- and two-year-old seedlings are mentioned in the previous description [48]. The 2-month-old seedlings were used for stress- and induction-related expression profile analysis. For drought and high-salinity treatment, 20% PEG6000 and 200 mM NaCl solution were used to treat the seedlings. For hormone induction, the seedlings were sprayed with ABA at a concentration of 100 mM. The above samples were obtained at 0 h, 1 h, 3 h, 6 h, and 12 h after each stress treatment, and then stored at −80 °C until use. Tissue samples (stem, root, leaf, and flower) were collected from 2-year-old plants and used for RNA sequencing. The resulting data were normalized using TBtools v 0.58 [49] to generate a heatmap. Other samples were used for real-time quantitative PCR (RT-qPCR) analysis. 

The seeds of *Arabidopsis* were surface-sterilized and sown in 1/2 MS medium containing 2% sucrose. Plants were grown in growth chambers under long-day conditions (16 h light and 8 h darkness) at 22 °C. After ten days, different lines were transferred to the soil. In the experimental group, 4-week-old seedlings were treated for 15 days without watering. The control group was watered once every 3 d. After 15 days of treatment, the phenotypes of these three *Arabidopsis* plants were observed. Three replicates were conducted, and each replicate contained at least 9 seedlings.

Total RNA was extracted using the HiPure HP Plant RNA Mini Kit (Magen, Guangzhou, China). cDNA was synthesized with Prime Script RT Master Mix (Takara, Dalian, China). The acquired cDNA was diluted 10-fold. RT-qPCR was carried out on the Roche Light Cycler 96 system (Roche Diagnostics GmbH, Mannheim, Germany) using SYBR^®^ qPCR Master Mix (Takara, Dalian, China) [50]. Technical and biological replicates were repeated three times. All of the primers checked by Primer-BLAST (Appendix A) were designed using GenScript. The relative expression levels of *HpbZIPs* were determined via the 2^−ΔΔCt^ method [51]. *HpActin2* (MK054303) was screened out as the internal reference [52].

### 4.4. Subcellular Localization of the HpbZIP69

The whole-length coding nucleotide sequences of *HpbZIP69* were cloned into the vector pDONR207 using a BP reaction of the Gateway (Invitrogen, Carlsbad, CA, USA) [53]. The pDONR207-*HpbZIP69* was recombined into the destination vector pEarleyGate103 (CD3-685) to determine the subcellular localization expression. All primers used in the experiment are shown in Appendix A. Meanwhile, pEarleyGate103 was used as a positive control. The vectors were bombarded with the gene gun using the method described previously [54]. The images were shot at 475 nm from living onion cells, recorded using a Leica DM6000B microscope (Leica, Wetzlar, Germany) after incubation on solid MS medium at 28 °C for 24 h in the dark.

### 4.5. Transcriptional Activation and Y1H Assay

To experiment the transcriptional activity of HpbZIP69, the sequence with a full-length coding region (374 amino acids) and fragmentary sequences containing 1–298 amino acids (*N*-terminal), 1–319 amino acids, 229–374 amino acids, and 320–374 amino acids (*C*-terminal) were inserted into pGBKT7, separately. The recombinant vectors and the empty pGBKT7 (negative control) were transferred into the AH109 yeast-competent cells (Weidi, Shanghai, China). The transformants were grown on SD/-Trp and SD/-Trp/-Ade/-His/X-α-gal deficiency media. Then, the transcriptional activities were tested according to their growth status at 29 °C for three days in darkness. Since the promoter sequence of *HpASMT2* contains four G-box elements, it was constructed on the pHis2 vector as a reporter vector. The pGADT7-*HpbZIP69* was constructed as an effector vector. All recombinant plasmids were transformed into yeast-competent Y187 cells (Weidi, Shanghai, China) and screened on SD/-Trp/-Leu medium at 29 °C for 48–72 h. Then, the surviving colonies were transferred to SD/-Trp/-Leu/-His media with and without 60 mM 3-AT (3-amino-1,2,4-triazole, Coolaber, Beijing, China). The variety of p53His and pGADT7-p53 vectors served as positive controls, while the combination of p53His and pGADT7 served as negative controls. The interaction between HpbZIP69 and G-box motifs was valued based on the transformants’ growth status. The primers used in the experiment are listed in Appendix A.

### 4.6. Overexpressing HpbZIP69 in Arabidopsis

The coding sequence of *HpbZIP69* containing attB and attP sites was inserted into pDONR207 by a BP reaction, according to the principle of Gateway (Invitrogen, Carlsbad, CA, USA). Then, pEarleyGate202-*HpbZIP69* was constructed by an LR reaction. The constructs, 35S:: HpbZIP69 and empty pEarleyGate202, were transferred into *Agrobacterium tumefaciens* strain GV3101 (Weidi, Shanghai, China). Transgenic *Arabidopsis* lines were obtained by an *Agrobacterium*-mediated transformation method [55] and screening of corresponding antibiotics. They were detected and identified at the DNA level and RNA level, respectively. Homozygous T3 transgenic lines were detected by semi-quantitative PCR as described previously [54], and the seedlings were used for subsequent analysis. The primer sequences can be found in Appendix A.

### 4.7. Physiological Index Measurement

One-month-old WT and OE field plants were analyzed for drought resistance. Malondialdehyde (MDA) and hydrogen peroxide (H_2_O_2_) concentrations were measured using the MDA Assay Kit and H_2_O_2_ Assay Kit, respectively (Solarbio, Beijing, China). The superoxide dismutase (SOD) level was measured using an SOD Assay Kit (Solarbio, Beijing, China) and then examined for values using a Multiskan FC microplate photometer (Thermo Fisher, Waltham, MA, USA), with three biological and three technical replicates. ANOVA was used for statistical analysis, and *p*-values < 0.05 were considered statistically significant.

## 5. Conclusions

In conclusion, the bZIP TF family is of vital importance for the growth and development of *H. perforatum*. Based on the phylogenetic tree, gene structure analyses, stress- and hormone-related cis-acting elements, and expression patterns in different tissues and under abiotic stresses, *H. perforatum* bZIP TFs were analyzed by bioinformatics and qRT-PCR. Most *H. perforatum* bZIP TFs may be involved in numerous abiotic stress responses. Overexpression of *HpbZIP69* enhanced drought tolerance in *Arabidopsis*. However, the function of *HpbZIP69* in *H. perforatum* remains to be proven experimentally in further research.

## Figures and Tables

**Figure 1 ijms-24-14238-f001:**
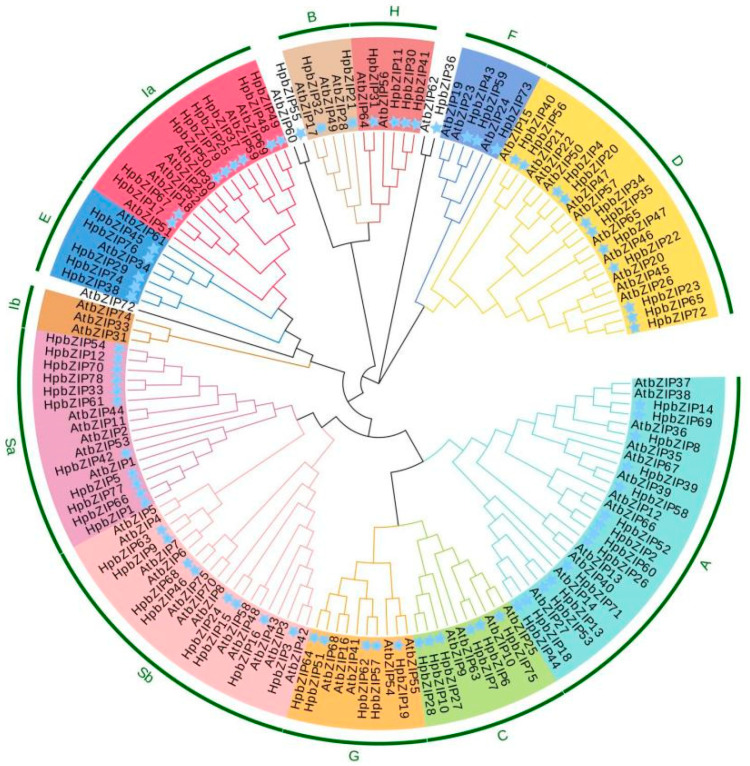
Phylogenetic analysis and subgroup classification of HpbZIP proteins: The phylogenetic tree of bZIP proteins was constructed based on the homologous proteins in *Arabidopsis*. The 79 HpbZIP proteins and 75 AtbZIP proteins were categorized into ten groups (A–I and S). The blue stars represent the bZIP protein of *H. perforatum*. The neighbor-joining tree was constructed using MEGA 7.0 software.

**Figure 2 ijms-24-14238-f002:**
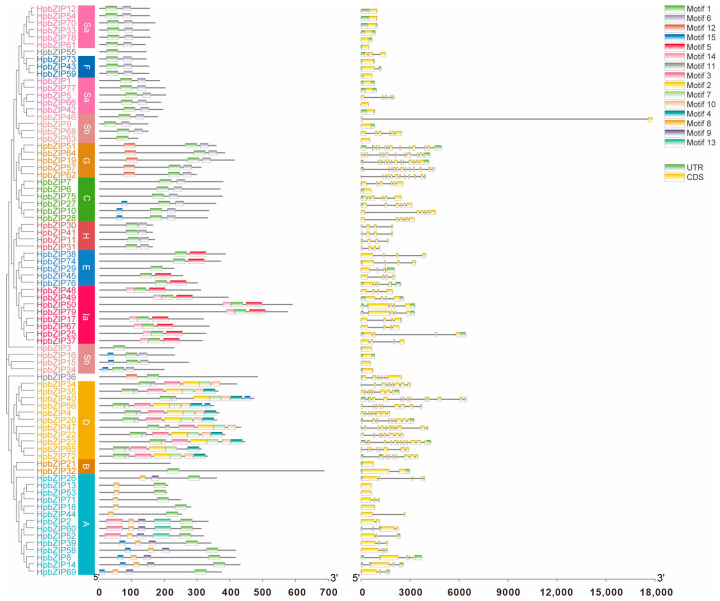
Phylogenetic trees, conserved motifs, and gene structures of *HpbZIP* genes. In the left part, different subgroups are indicated with various background colors. In the middle part, boxes represent motif distributions. In the right part, green boxes, grey lines, and yellow boxes represent exons, introns, and UTRs, respectively. The results were displayed using TB tools.

**Figure 3 ijms-24-14238-f003:**
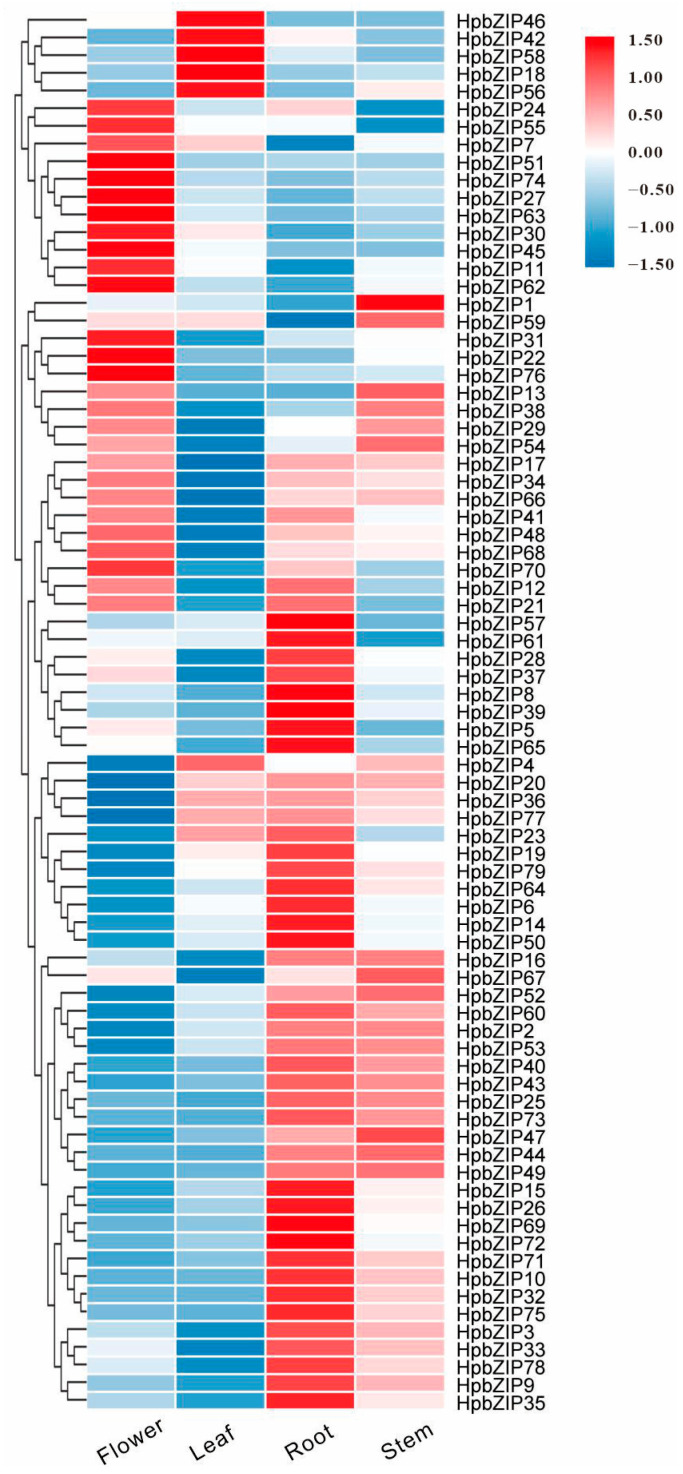
Expression profiles of *HpbZIP* genes: Hierarchical clustering of expression profiles of 79 *HpbZIP* genes in different tissues (root, stem, leaf, and flower).

**Figure 4 ijms-24-14238-f004:**
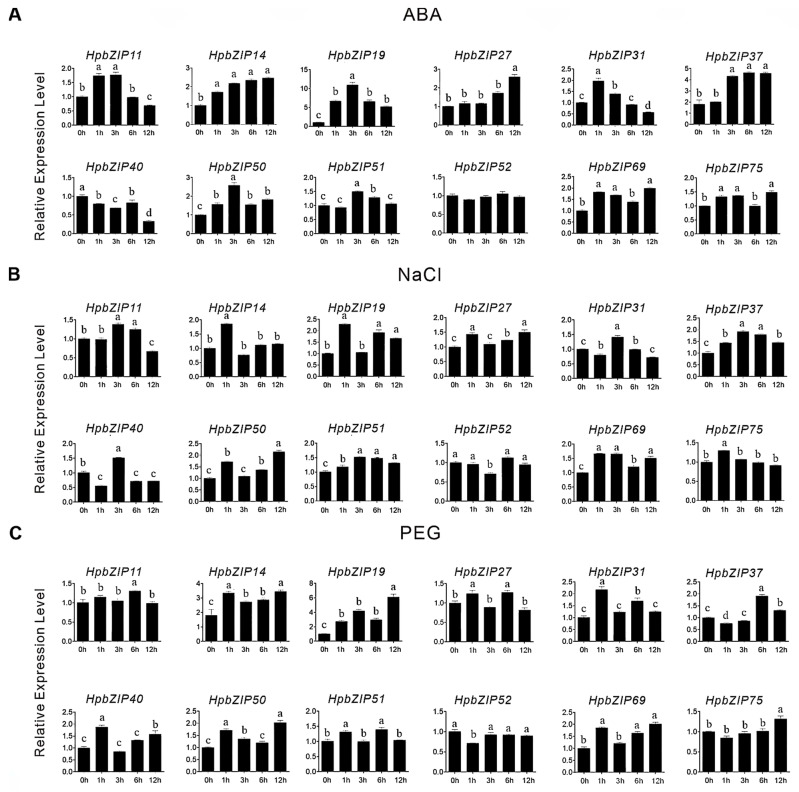
Expression analysis of 12 representative *HpbZIP* genes under 100 mM ABA induction (**A**), 200 mM NaCl (**B**), and 20% PEG6000 (**C**) treatments. Data were normalized to *HpActin2*, calculated with the equation 2^−ΔΔCt^, and vertical bars indicate standard deviation. Each treatment contains three biological replicates. Data represent the means from three biological and three technical replicates, and error bars indicate ± SD. Different letters indicate significant differences (*p* < 0.05) tested by one-way ANOVA with the Tukey’s multiple comparison test).

**Figure 5 ijms-24-14238-f005:**
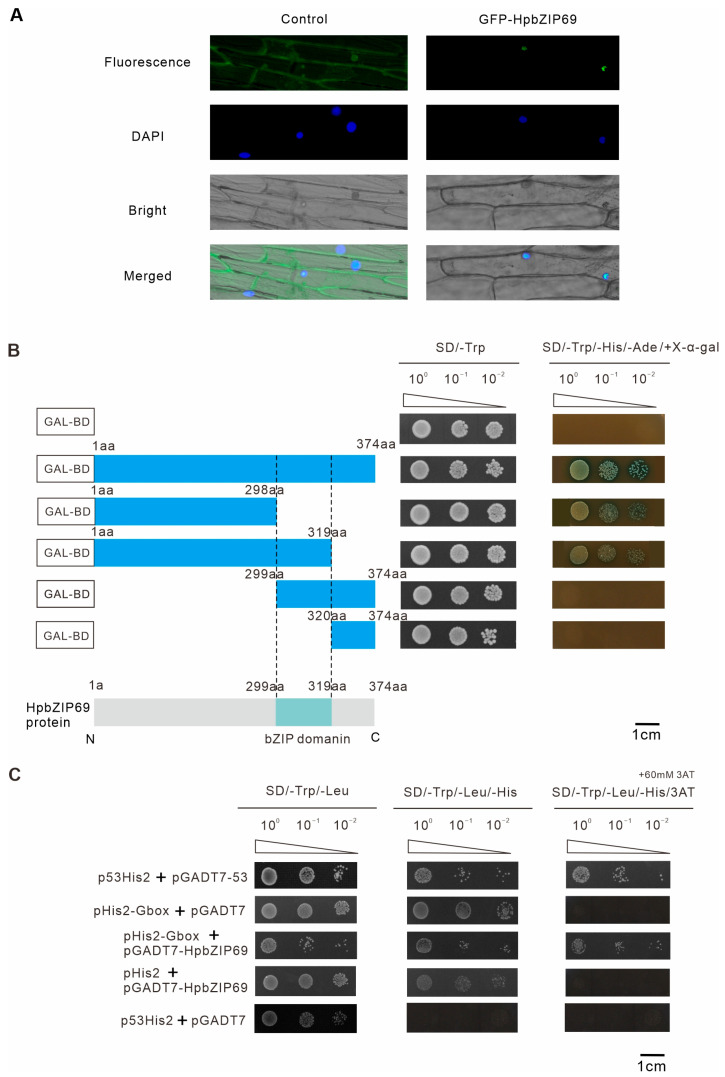
*HpbZIP69* is a representative bZIP transcription factor: (**A**) The subcellular location of HpbZIP69. Scale bars: 30 µm. (**B**) Transactivation activity test in the yeast GAL4 system and the protein structure of HpbZIP69. (**C**) Yeast one-hybrid assay revealing that HpbZIP69 can bind to the G-box motif.

**Figure 6 ijms-24-14238-f006:**
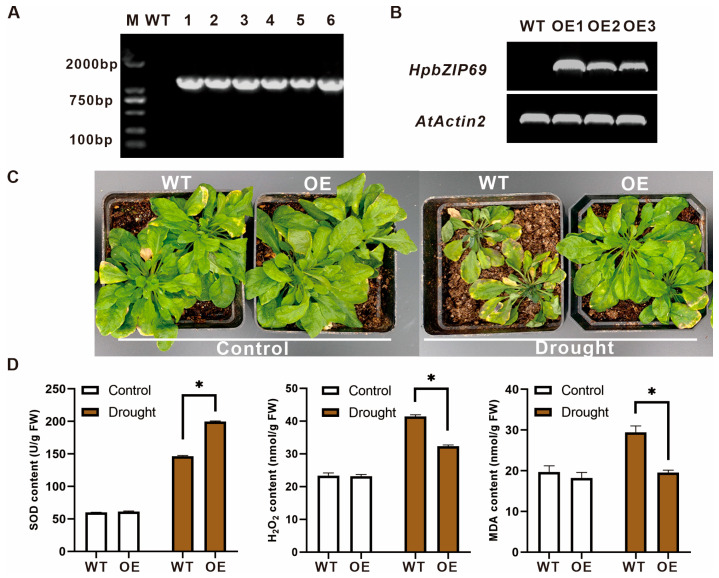
Function analysis of *HpbZIP69* under drought treatment in transgenic *Arabidopsis*: (**A**) PCR results of *HpbZIP69* genes. *HpbZIP69* cannot be cloned from WT lines, but it can be cloned from OE transgenic lines. (**B**) RT-PCR identification of positive transgenic *A. thaliana* lines. The *AtActin2* gene was used as an inner reference. The WT line was used as the negative control. (**C**) Observations of WT and OE seedlings under drought testing. (**D**) The analysis of SOD, H_2_O_2_, and MDA contents in WT and OE lines under control and drought conditions. Data are the mean ± SD of three independent biological replicates (*: *p* < 0.05; Two-way ANOVA test).

## Data Availability

Not applicable.

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
