# Peer review of "Systematic Identification and Functional Analysis of the Hypericum perforatum L. bZIP Gene Family Indicating That Overexpressed HpbZIP69 Enhances Drought Resistance"

_ijms, 2023, doi:10.3390/ijms241814238_

Round 1

Reviewer 1 Report

1. The general direction of the manuscript “Systematic identification and functional analysis of the Hypericum perforatum L. bZIP gene family indicating that overexpressed HpbZIP69 enhances drought resistance” is the search for promising targets for modulating plant resistance to abiotic stresses, such as drought, as well as the study of the molecular mechanisms of these targets. Methodologically, this work was done at a high level, corresponding to the journal. The conclusions are consistent with the evidence and arguments presented and do they address the main question posed. The references are appropriate.

There are some minor comments:

1.      Title, keywords and Abstract (as and all text) could be intense proofread.

2.      It is unclear from abstract, why exactly HpbZIP69 was overexpressed

3.      Description of ABF3 (AtbZIP37) and ABF4/AREB2 (AtbZIP38) is presented only in discussion. However, authors selected HpbZIP69 for detailed analysis according to their homology. It will be better to include description of the ABF3 (AtbZIP37) and ABF4/AREB2 (AtbZIP38) in Introduction as target homologues for HpbZIP genes.

4.      Line 202 – there are not Fig 3A or 3B; Please, correct. Correct figure captures (Fig 3); what method was used?

5.      Line 204: Are the 1.5-times fold changes enough for conclusion “implying that gene expression might 205 be development-specific”?

6.      Figure 4: Please increase gene names (Unreadable). Please, add in the figure captures NaCl, ABA and PEG concentrations.

7.      Line 218: expression of HpbZIP19 under ABA, NaCl and PEG treatment was more induced (up to 15 times) than HpbZIP69 (less than two times). HpbZIP19 is more perspective for analysis. So, see point 3. Without description of target (closest homolog of ABF3 (AtbZIP37) and ABF4/AREB2 (AtbZIP38)) further studies lose their relationship with previous ones.

8.      If authors have more than three OE-lines, why only one was used for drought stress? SemiQ analysis is not sufficient to determine the level of expression. If other OE-lines did not show drought resistance, this can be brought in the supplemental and discussed. Perhaps HpbZIP19 would be more interesting as an object for further research. Also, it would be good to add a discussion of the closest HpbZIP19 homologues, what is known about their function.

9.      Please, italize plant names through text.

Best regards

Author Response

Thank you so much for your affirmation of our work and the specific comments and suggestions. Here is my reply. Please review:

  1. Thank you for your comments. We have corrected and proofread the all text including the title, keywords, and abstract. Please see corresponding sections.
  2. Thank you for your comments. We have added related content of HpbZIP69in the abstract. Please refer to Line 26-28.
  3. Thank you for your comments. We have added relevant content in introduction. Please refer to Line 76-81, and Line 107-112.
  4. Sorry for the mistake. We have correct it (Line 225) and the spicific method used to draw the heatmap has been added to the text, please refer to Line 244-245and 411-415.
  5. Thank you for your comments. Please see Line 225-227.
  6. Thank you for your comments. We redrawed Figure 4 and labeled the name of the genes. Additional, concentrations of NaCl, ABA,and PEG were added in the legend. Please see Line 248-249.
  7. Thank you for your comments. Based on previous studies, it was difficult to find relevant reports about HpbZIP19in plant stress resistance, so we did not choose 19 as our research object. As for other description of target (closest homolog of ABF3 (AtbZIP37) and ABF4/AREB2 (AtbZIP38)) was added in the text. Please refer to Line 76-81.
  8. Thank you for your comments.Due to the heterologous overexpression of the HpBZIP69gene of H. perforatum in Arabidopsis, it is impossible for Arabidopsis Col-0 to express HpbZIP69. Therefore, there is no control group and qRT-PCR cannot be used to study the expression level. Referring to published literature, semi quantitative PCR was found to be the most common and direct method for detecting the expression of target genes. We obtained three overexpressed Arabidopsis lines and found in the preliminary experiment that OE1 showed more significant drought resistance, which is consistent with our semi quantitative PCR results. So we directly used OE1 for the next experiment. The suggestions given by the reviewer are very accurate, and in future works, we will use multiple strains for functional exploration, which will result in more rigorous experimental data. We also noticed the HpbZIP69 mentioned by the reviewer and found that its expression pattern showed a strong response to various processes. However, in evolutionary and structure analysis, it was found that its homologous gene AbZIP54/55 had no reported function in the literature, so there was no in-depth research. Thank you for the reviewer's reminder. In our later work, we will list HpbZIP69 as a candidate gene for research.
  1. Thank you for your comments. We have revised the format of all plant names to italics in text.

Reviewer 2 Report

Manuscript "Systematic identification and functional analysis of the Hypericum perforatum L. bZIP gene family indicating that overexpressed HpbZIP69 enhances drought resistance" is very interesting.

General comments:
Authors analyzed the number of genes, classification, and stress-produced expression modes of bZIP TF family members in H. perforatum were systematically analyzed based on whole genome data. Authors analyzed the phenotype and RNA-seq information of HpbZIP69 transgenic Arabidopsis to investigated the characteristics and molecular functions of HpbZIP69 under drought stress.

Detailed comments:
Figure 1: provide the measure used to calculate similarity and the method of dendrogram construction.
Figure 2: provide the measure used to calculate similarity and the method of dendrogram construction.
Figure 3: provide the measure used to calculate similarity and the method of dendrogram construction.
Figure 4: should be supplemented with LSD values for comparing average values.
Figure 6: the data shown indicate a two-factor experiment. Appropriate statistical methods should be used to evaluate such an experiment. In this case, it must be a two-way ANOVA.
The description of the statistical methodology is very poor. It needs to be supplemented with, among other things, a versification of the assumptions required to apply the analysis of variance. Methods appropriate to the assumed experiment should be used.

My suggestions:
Line 103: "Arabidopsis thaliana" - italic.
Line 585: "Arabidopsis thaliana" - italic.

Paper needs major revision.

Author Response

Thank you so much for your affirmation of our work and the specific comments and suggestions. Here is my reply. Please review:

  1. Thank you for your comments. Please refer to Line 150-151and 389-393.
  2. Please refer to Line 195-197and 377-385.
  3. Please refer to Line 244-245and 411-414.
  4. Thank you for your comments. We have supplemented LSD values for comparing average values in Figure 4.
  5. Sorry for the mistakes, the Figure 6 uploaded is the uncorrected version, we have re-uploaded Figure 6, please check the result 2.7 section.
  6. Thank you for your comments. We have revised the format of all plant names and gene names to italics in text.
  7. Thank you for your comments. We have revised the format of all plant namesand gene names to italics in text.

Round 2

Reviewer 2 Report

Figures 1, 2 and 3: The measure used to calculate the similarity is still not specified.

Figure 4: In response to a comment about the lack of LSD values, the authors wrote that they had added these values. Unfortunately, the values are still not there. Why this lie?

Still Figure 6: the data shown indicate a two-factor experiment. Appropriate statistical methods should be used to evaluate such an experiment. In this case, it must be a two-way ANOVA.

The description of the statistical methodology is still very poor. It needs to be supplemented with, among other things, a versification of the assumptions required to apply the analysis of variance. Methods appropriate to the assumed experiment should be used. The authors have done nothing to improve this.

Author Response

Many bioinformatics analyses of transcription factor gene families are conducted using well-known programs or software as well as default parameters, so there is not much description. We are very sorry for any statistical omissions and would like to thank the reviewer for their professional feedback. We have made further modifications to the method section, please review.

Figures 1, 2 and 3: The measure used to calculate the similarity is still not specified.

We apologize for not correctly understanding the reviewer's comments. We provided a detailed description of the method for constructing evolutionary trees, but we are not aware that you are referring to specific similarity calculations.

Evolutionary tree analysis requires two steps: first, multiple sequence alignment, and secondly, tree construction. The Neighbour-Joining method we use evolves a paired evolutionary distance matrix between given sequences, and the paired distance is usually obtained from sequence alignment algorithms, such as ClustalW/X. The comparison score can be used as an estimate of the evolutionary distance between sequences. Here, ClustalW algorithm with gap opening penalty 10.00 and gap extention penalty 0.20 were used. Please refer to Line 383-389.

Figure 4: In response to a comment about the lack of LSD values, the authors wrote that they had added these values. Unfortunately, the values are still not there.

We sincerely apologize again for not correctly understanding the reviewer's comments.

We used Graphpad Prism 8 for statistical analysis of the data. After One-way ANOVA analysis, we selected pairwise comparisons mean from multiple comparisons using the software's recommended Paired t-Test method (Line 245-247). We also referred to other literature recommendations on statistical methods for comparing gene expression levels and selected Paired t-Test. Here, I take the expression level data of Hpbzip11 induced by ABA as an example:

Tukey's Multiple Comparison

Mean Difference

Significant

Summary

Adjusted P-value

0 h vs. 1 h

-0.6592

Yes

****

<0.0001

0 h vs. 3 h

-0.7163

Yes

***

0.0001

0 h vs. 6 h

-0.05247

No

ns

0.8460

0 h vs. 12 h

0.4665

Yes

**

0.0013

1 h vs. 3 h

-0.05710

No

ns

0.0751

1 h vs. 6 h

0.6068

Yes

*

0.0324

1 h vs. 12 h

1.126

Yes

***

0.0001

3 h vs. 6 h

0.6639

Yes

*

0.0235

3 h vs. 12 h

1.183

Yes

***

0.0004

6 h vs. 12 h

0.5190

Yes

*

0.0308

Different letters indicate significant differences (P < 0.05)

We have not addressed such issues in previous gene family analysis, if reviewer need us to provide supplementary materials with values and statistical data for each gene under different treatments. Please let us know, we will attach

Figure 6: the data shown indicate a two-factor experiment. Appropriate statistical methods should be used to evaluate such an experiment. In this case, it must be a two-way ANOVA.

Thank you very much for pointing out our mistake. Yes, there are two variables involved in this experiment: the drought treatment experiment between different plant Lines. So the statistical method must be two-way ANOVA. We carefully checked the calculation method at the time and it was two-way ANOVA. I have made corrections to the figure legend, please refer to Line 293.